# Biosynthesis of Smaller-Sized Platinum Nanoparticles Using the Leaf Extract of *Combretum erythrophyllum* and Its Antibacterial Activities

**DOI:** 10.3390/antibiotics10111275

**Published:** 2021-10-20

**Authors:** Olufunto T. Fanoro, Sundararajan Parani, Rodney Maluleke, Thabang C. Lebepe, Rajendran J. Varghese, Nande Mgedle, Vuyo Mavumengwana, Oluwatobi S. Oluwafemi

**Affiliations:** 1Department of Biotechnology, University of Johannesburg, Doornfontein, Johannesburg 2028, South Africa; jolufunto@gmail.com (O.T.F.); vuyom@uj.ac.za (V.M.); 2Centre for Nanomaterials Sciences Research, University of Johannesburg, Doornfontein, Johannesburg 2028, South Africa; sbarani416@gmail.com (S.P.); rodney.maluleke@gmail.com (R.M.); calvyn.tl@gmail.com (T.C.L.); josv3209@gmail.com (R.J.V.); nandemgedle@gmail.com (N.M.); 3Department of Chemical Sciences (Formerly Applied Chemistry), University of Johannesburg, Doornfontein, Johannesburg 2028, South Africa

**Keywords:** *Combretum erythrophyllum*, platinum nanoparticles, green synthesis, antibacterial, *Klebsiella oxytoca*, *Klebsiella aerogenes*

## Abstract

Nanobiotechnology is a promising field in the development of safe antibiotics to combat the increasing trend of antibiotic resistance. Nature is a vast reservoir for green materials used in the synthesis of non-toxic and environmentally friendly nano-antibiotics. We present for the first time a facile, green, cost-effective, plant-mediated synthesis of platinum nanoparticles (PtNPs) using the extract of *Combretum erythrophyllum* (CE) plant leaves. The extract of CE served as both a bio-reductant and a stabilizing agent. The as-synthesized PtNPs were characterized using ultraviolet-visible (UV–Vis) absorption spectroscopy, high-resolution transmission electron microscopy (HR-TEM), Fourier transform infrared spectroscopy (FTIR), X-ray diffraction (XRD), and dynamic light scattering (DLS) techniques. The HR-TEM image confirmed that the PtNPs are ultrasmall, spherical, and well dispersed with an average particle diameter of 1.04 ± 0.26 nm. The PtNPs showed strong antibacterial activities against pathogenic Gram-positive *Staphylococcus epidermidis* (ATCC 14990) at a minimum inhibitory concentration (MIC) of 3.125 µg/mL and Gram-negative *Klebsiella oxytoca* (ATCC 8724) and *Klebsiella aerogenes* (ATCC 27853) at an MIC value of 1.56 µg/mL. The CE-stabilized PtNPs was mostly effective in *Klebsiella* species that are causative organisms in nosocomial infections.

## 1. Introduction

Nanobiotechnology, an emerging field of nanoscience, has advanced the development of green practices in producing bio-inspired nanoparticles (NPs). It involves using biomolecules in the production of nanomaterials and nano-based systems. These biomolecules account for the variation in the size, shape, wavelength, surface area, and functionalization of these nanoparticles [1,2,3]. Noble metal nanoparticles (MNPs) such as platinum, gold, and silver have been used in diverse applications such as pharmaceutics, gas sensors, electronics, medicine, diagnosis, fuel cells, sensing, drug, gene delivery, etc. This is due to their excellent intrinsic optical, electronic, and physicochemical properties, which offer them functional capabilities different from the bulk materials [2,4], giving “Ockham’s razor” to applied nanomedicine because of the flexibility in their multi-functionalization in bio-applications [5]. Among the noble MNPs, platinum nanoparticles (PtNPs) possess inherent and distinct properties such as high surface area, resistance to corrosion, and biocidal effects [6]. Several physical and chemical methods have been used in the synthesis of PtNPs. However, these processes are not completely green due to the use of high temperatures and toxic reagents. In addition, large-scale synthesis is not easy to achieve with such methods [7]. To proffer alternatives to these procedures, a benign and environmentally friendly approach has been used by prospecting microorganisms [8], plant extract [9,10], and polysaccharides [11] for the synthesis of PtNPs. Amongst these biological entities, the use of plant extracts (PE) has proved to be an excellent choice because they are readily available, simple, cost-effective, eco-friendly, and highly bio-compatible for biomedical applications [12]. Furthermore, PE contains diverse phytochemicals or metabolites such as carbohydrates, alkaloids, terpenes, phenols, tannins, lipids, quinones, reductases, proteins, flavonoids, vitamins, etc. [13,14,15]. Therefore, they can serve as both reducing and stabilizing agents, thus eliminating the use of hazardous chemical reagents in the synthesis of MNPs [16]. These phytochemicals or metabolites can prevent the aggregation of nanoparticles, reduce generic toxicity, and increase their bio-assimilation [4,17,18]. Several plant, leaf, flower, and seed extracts of *Taraxacum laevigatum* [9], *Garcinia mangostana* L. [19], *Xanthium strumarium* [20], *Punica granatum* [2], *Nigella sativa L*. [6], etc. have been used in the biogenic synthesis of PtNPs. Despite these, it is imperative to explore more plant sources due to their vast array of phytochemicals.

*Combretum erythrophyllum* (CE) is a native plant in South Africa belonging to the medicinal family of Combretaceae. It grows independently and usually serves as a shade or an ornamental plant [4,21]. Different extracts of the CE plant’s leaves, bark, and seeds have been used customarily for medicinal purposes due to their richness in secondary metabolites [22]. Prior studies have shown that its leaf and extract contain flavonoids, alkaloids, phenolics, carbohydrates, proteins, and essential oils [20,22], which are excellent bio-reductants. Despite these excellent properties, there has been no known research on the use of *C.*
*erythrophyllum* extract in the green synthesis of PtNPs or extracts from the Combretaceae family, as it has been reported for other MNPs such as silver and gold. Furthermore, as far as the authors know, there has been no report on the antibiotic properties of PtNPs synthesized using the extract of *C. erythrophyllum*. Herein, we report for the first time the synthesis of PtNPs using the aqueous extract of CE leaves as both reducing and stabilizing agents. The formation of the as-synthesized PtNPs was confirmed by using Ultraviolet-Visible (UV–Vis) absorption spectroscopy. The morphological and structural evaluation was carried out by using high-resolution transmission electron microscopy (HR-TEM). The biological activity of the as-synthesized PtNPs was evaluated using minimum inhibition concentration (MIC) for its antibacterial activity against Gram-positive and Gram-negative bacteria. The results showed that the CE-PtNPs exhibited selective antibacterial activity toward specific pathogenic *Klebsiella* species, which could serve as a means for controlling infectious diseases.

## 2. Results and Discussion

### 2.1. UV-Visible Spectra

The growth of biosynthesized PtNPs using C. *erythrophyllum extract* was monitored by observing the change in the color of the reaction mixture. As the reaction progressed, the color changed from light yellow to brown and finally darkish brown. This observed color change shows the conversion of the Pt (IV) to Pt (II) and finally to Pt (0), indicating the formation of PtNPs. The observed color change can be attributed to the surface plasmon response of MNPs due to the vibration of free electrons on its surface. The formation of PtNPs at 24 h reaction time was confirmed by UV-Vis spectroscopy. The absorption spectra of the precursor platinum (Pt) salt showed a plasmon peak at 262 nm as a result of the dissociation and formation of 2H^+^ and PtCl_6_^2−^ in water [19]. Nevertheless, after the reduction by the CE extract, the plasmon peak of the Pt salt disappeared, thus signifying a total reduction of the Pt salt to a zero-valent platinum (Figure 1). A similar result was reported for the biosynthesis of PtNPs using the extract of *Psidium guajava* [23].

### 2.2. XRD Analysis

The XRD pattern of the CE-PtNPs at 24 h is shown in Figure 2. The diffraction pattern shows peaks at 2θ values of 39.99°, 46.49°, 67.70°, and 81.49°, corresponding to the (111), (200), (220), and (311) crystallographic planes of the face-centered cubic (*fcc*) with the lattice parameter a = 3.92 Å, which is in accordance with JCPDs-ICDD Card no: 04-802 [24]. Similar results have been reported in the biosynthesis of PtNPs using *Punica granatum* crusts extract [2]. The high intensity of the peak at 39.99° shows that the nanoparticles were predominantly oriented toward the (111) plane. The width of the (111) peak was used to calculate the average crystallite size by using the Scherrer equation [25]. The calculated average size was 1.83 nm, which is in agreement with the size obtained from the HR-TEM.

### 2.3. TEM Analysis

Figure 3 shows the TEM micrograph of the as-synthesized PtNPs using the CE extract as a bio-reductant. As shown in Figure 3a, the as-prepared PtNPs are small and mostly spherical. The presence of lattice fringes in the high-resolution image indicates the crystallinity of the PtNPs. Figure 3b shows the correlation of the calculated lattice (d) spacing values with the XRD patterns. Furthermore, the selected area electron diffraction (SAED) image shows the ring patterns accompanied by the single spots in a ring (Figure 3b inset), which are in agreement with the XRD patterns. The particle size distribution that was obtained from the TEM micrograph is shown in Figure 4a. The particle size ranged between 0.2 and 1.8 nm with an average particle diameter of 1.04 ± 0.26 nm. In Figure 4b, the energy-dispersive X-ray spectrum (EDX) showed the occurrence of Pt and elements such as calcium, oxygen, and potassium, which come from CE extract, showing its richness in minerals. The Cu is a result of the copper grid used for the analysis. The PtNPs-CE showed a negative zeta potential of −34.1 mV, signifying sufficient surface charge for electrostatic and colloidal stability in biological systems [4].

### 2.4. FTIR Analysis

FTIR spectroscopy was used to investigate the surface chemistry of the as-synthesized PtNPs and confirm the functional groups of the biomolecules involved in reducing and capping the as-synthesized PtNPs. The FTIR spectra of both the CE and the as-synthesized PtNPs are shown in Figure 5, with common absorption bands presented in Table 1. The major absorption bands in the PtNPs are 3283 cm^−1^, 2920 cm^−1^, 2850 cm^−1^, 1708 cm^−1^, and 1620 cm^−1^. The peak at 3283 cm^−1^ correlates to the O-H stretching (intramolecular bonding) of the hydroxyl group found in polyphenols; peaks at 2920 cm^−1^ and 2850 cm^−1^ are attributed to the asymmetric stretch of C-H group of alkanes. The peak at 1708 cm^−1^ is attributed to the C=O stretching found in typical flavonoids or flavones. The peak at 1620 cm^−1^ correlates to the carboxylate anion stretching (-COO-) of proteins and amino acids. The shift observed in the wavenumber of -OH, C-H, C=O, and -COO- stretching of PtNPs compared to the CE extract shows possible coordination between the PtNPs and the CE extract [26]. Thus, it is believed that the proteins, flavonoids, amino acids, polyphenols, and carbohydrates biomolecules present in CE served as both bio-reductant and stabilizing agents.

### 2.5. Antibacterial Activity

The as-synthesized PtNPs were evaluated for antibacterial activities against diverse pathogenic Gram-positive (+ve) and Gram-negative (−ve) bacteria. The as-synthesized PtNPs exhibited inhibitory activity against the pathogenic bacteria at different concentrations, as shown in Appendix A. Figure 6 shows a graphical presentation of the MIC values of the PtNPs tested against the listed bacterial strains. Significantly, the as-synthesized PtNPs showed selectivity toward Gram-negative *Klebsiella* species of *Klebsiella aerogenes* and *Klebsiella oxytoca* at a very low MIC value of 1.56 μg/mL. *Staphylococcus epidermidis* and *Proteus mirabilis* showed susceptibility to the CE capped PtNPs but at a higher MIC value of 3.125 μg/mL. *Enterococcus faecalis*, *Bacillus subtilis*, and *Klebsiella pneumoniae* were susceptible at a much higher concentration, ranging from 500 to 2000 μg/mL. The PtNPs showed no inhibitory effect against *Escherichia coli*, *Staphylococcus aureus, Proteus vulgaris,* and *Mycobacterium smegmatis*. A preliminary study evaluating the antibacterial effect of the CE extract showed no effective inhibitory properties compared to the PtNPs (Appendix A). The observed low MIC value observed for the *Klebsiella* species of Ka and Ko could be due to oxidative stress as a result of reactive oxygen species (ROS) that are produced via a redox process. This eventually causes membrane and DNA damage that culminates in cell death. Cell death via ROS is often more potent [27]. A higher MIC value observed in *Klebsiella pneumoniae* can be attributed to the fact that it is one of the most virulent and resistant species of the *Klebsiella* genus. Such virulence factors are the production of hyper capsules, siderophores such as salmochelin and aerobactin, genetic codes for allantoin metabolism, and fimbriae [28,29]. All these are believed to contribute to the pathogenicity of *K. pneumoniae*, which possibly makes it less susceptible or resistant to ROS [29], and thus, a higher MIC value was obtained compared to the other *Klebsiella* species. Another plausible reason for the higher MIC values could be metal ion release. The metal ion interacts with the amine and carboxylic groups of proteins and nucleic acids to cause cell death. This is a less potent and slower mechanism, and thus, a larger amount of MNPs are required for cell death [27,30,31].

## 3. Materials and Methods

### 3.1. Materials

Analytical grade hexachloroplatinic (IV) acid hydrate (H_2_PtCl_6_∙xH_2_O) and Mueller–Hinton agar and broth were procured from Sigma-Aldrich, South Africa. One mg/mL streptomycin was purchased from Sigma Aldrich, St. Gallen, Switzerland (BCBP5897V). All aqueous solutions were prepared using deionized water.

### 3.2. Preparation of the Plant Extract

Healthy CE leaves were obtained from the Water Sisulu National Botanical Garden at Roodepoort, Johannesburg. The leaves were cleaned and afterwards dried under ambient conditions. Five grams of the dried leaves were mixed with 100 mL of deionized water and heated at 90 °C for 1 h. The mixture was filtered using a Whatman filter paper, and the filtrate was used for the synthesis.

### 3.3. Green synthesis of PtNPs

First, 10 mL of the extract from the CE leaf was added to 50 mL of 1 mM H_2_PtCl_6_∙xH_2_O at 85 °C under reflux at a stirring rate of 750 rpm for 24 h.

### 3.4. Characterization

A Perkin Elmer Lambda 25 UV-Vis spectrophotometer was used for the absorption measurement. The absorption spectra were recorded in the range of 200–700 nm. The shape and size of the as-synthesized PtNPs were determined by transmission electron microscopy (TEM) using JEOL JEM-2100 at an acceleration voltage of 200 kV. The zeta potential analysis of the PtNPs was studied at 25 °C using Malvern Panalytical Zetasizer Nano ZS based on the Smoluchowski model. The surface chemistry of the sample was investigated using Perkin Elmer Spectrum Two FTIR spectroscopy over the range of 4000–400 cm^−1^. X-ray diffraction (XRD) studies were conducted with monochromatic Cu- Kα1 radiation (λ = 1.54 Å) at the diffraction angle range of 10° and 90° using a Bruker D8 Advance X-ray diffractometer.

### 3.5. Antibacterial Activity of PtNPs

The microdilution technique was used in the evaluation of the MIC. Fresh bacterial growths of the listed pathogenic bacteria strains (*Staphylococcus epidermidis* (Se) (ATCC14990), Proteus *mirabilis* (Pm), (ATCC 7002), *Escherichia coli* (Ec) (ATCC 25922), *Enterococcus faecalis* (Ef) (ATCC 13047), *Bacillus subtilis* (Bs) (ATCC 19659), *Staphylococcus aureus (Sa) (ATCC 25923), Klebsiella pneumoniae (Kp) (ATCC 13822), Klebsiella oxytoca (Ko) (ATCC 8724), Mycobacterium smegmatis (Ms) (MC 2155), Bacillus cereus (Bc) (ATCC 10876)*, and *Klebsiella aerogenes* (Ka) (ATCC 27853)] were standardized to the 0.5 McFarland standards in Muller–Hilton broth, which was then used as inoculum. In 96-well plate containing 100 µL of 2000, 1000, 500, 250, 125, 62.5, 50, 25, 12.5, 6.25, 3.125 and 1.56 µg/mL of PtNPs; 100 µL of each inoculum were seeded in triplicate. The plates were grown overnight at 37 °C. Muller–Hilton broth (50% *v/v* in DMSO) was used as a negative control. Streptomycin (1 mg/mL) served as the positive control. Viable cells were confirmed in the presence of resazurin dye after 2 h incubation as they enzymatically reduced resazurin dye (blue color) to the resorufin that fluoresces pink. A pink color indicates bacterial growth. The smallest concentration that inhibited bacterial growth was recorded as the MIC, and the values were recorded for each bacteria strain.

## 4. Conclusions

A simple, cost-effective, green, and eco-friendly method was used to synthesize water-soluble CE stabilized PtNPs. The leaf extract *of Combretum erythrophyllum* served as the bio-reductant and stabilizing agent. An optical characterization by UV-Vis was used to confirm the formation of the PtNPs. XRD studies revealed the formation of PtNPs with a cubic crystal structure, while the FTIR study confirmed the role of the biomolecules of CE extract as a capping agent in the biosynthesis of the PtNPs. The CE capped PtNPs are spherical in shape with an average particle diameter of 1.04 ± 0.26 nm. The biosynthesized PtNPs showed selective inhibition toward Gram-negative Klebsiella species implicated in nosocomial infections. This study could serve as a starting point in exploring CE-capped PtNPs in nanomedicine for the treatment of nosocomial infectious diseases.

## Figures and Tables

**Figure 1 antibiotics-10-01275-f001:**
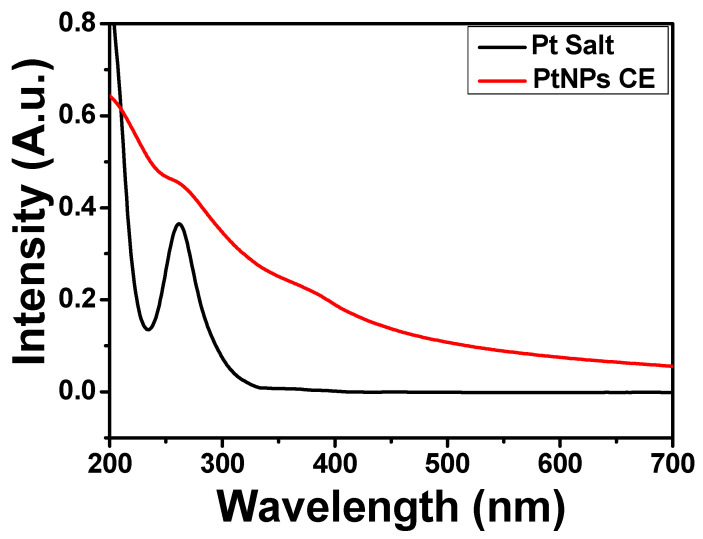
Absorbance spectra of platinum salt and PtNPs CE.

**Figure 2 antibiotics-10-01275-f002:**
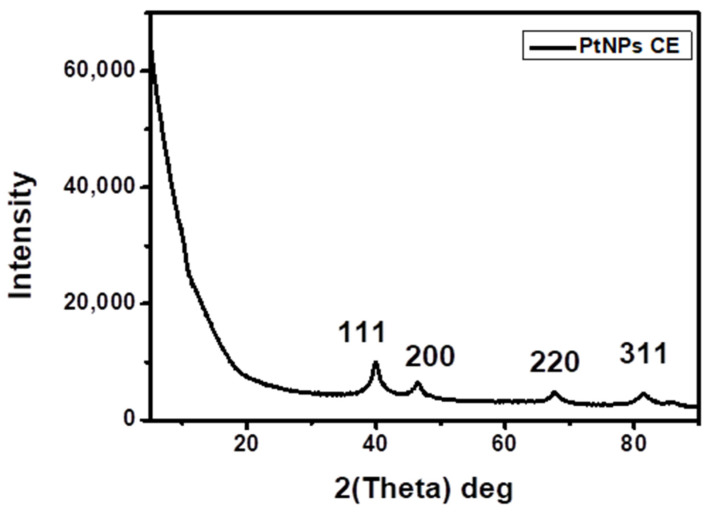
XRD patterns of the as-synthesized PtNPs CE.

**Figure 3 antibiotics-10-01275-f003:**
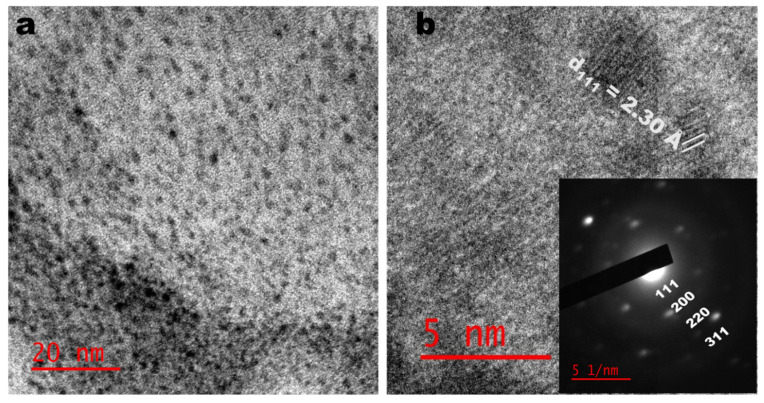
TEM (**a**) and HRTEM (**b**) images of CE synthesized PtNPs (Inset: SAED).

**Figure 4 antibiotics-10-01275-f004:**
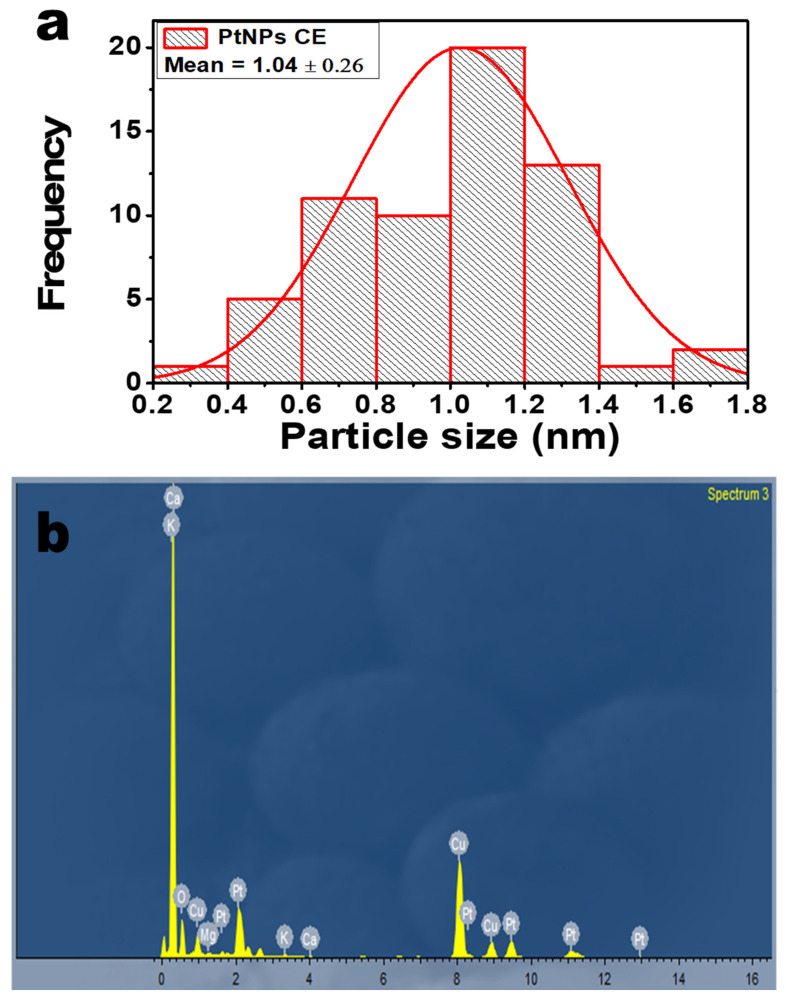
(**a**) Particle size distribution of PtNPs; (**b**) EDX spectra of PtNPs.

**Figure 5 antibiotics-10-01275-f005:**
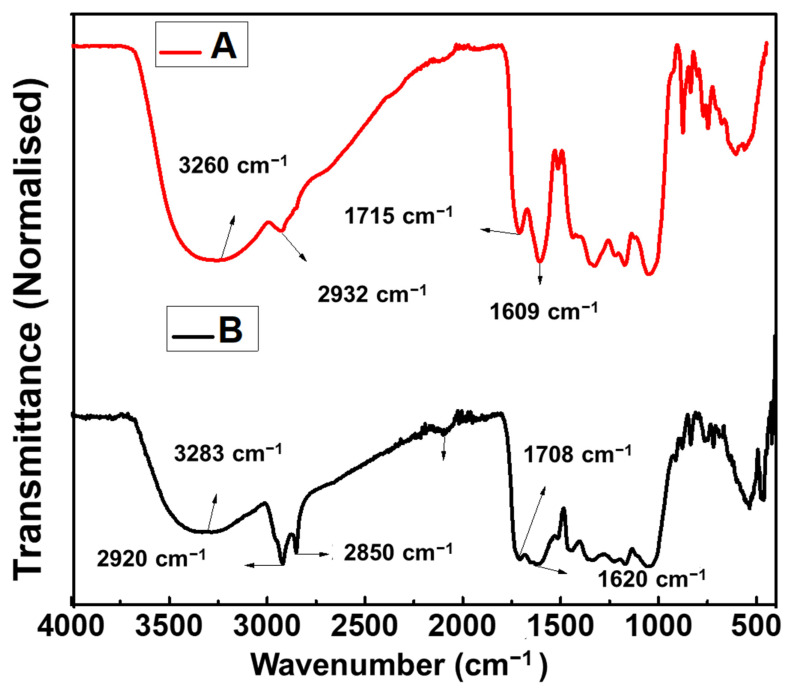
FTIR spectra of (**A**) CE extract, (**B**) PtNPs synthesized from CE extract.

**Figure 6 antibiotics-10-01275-f006:**
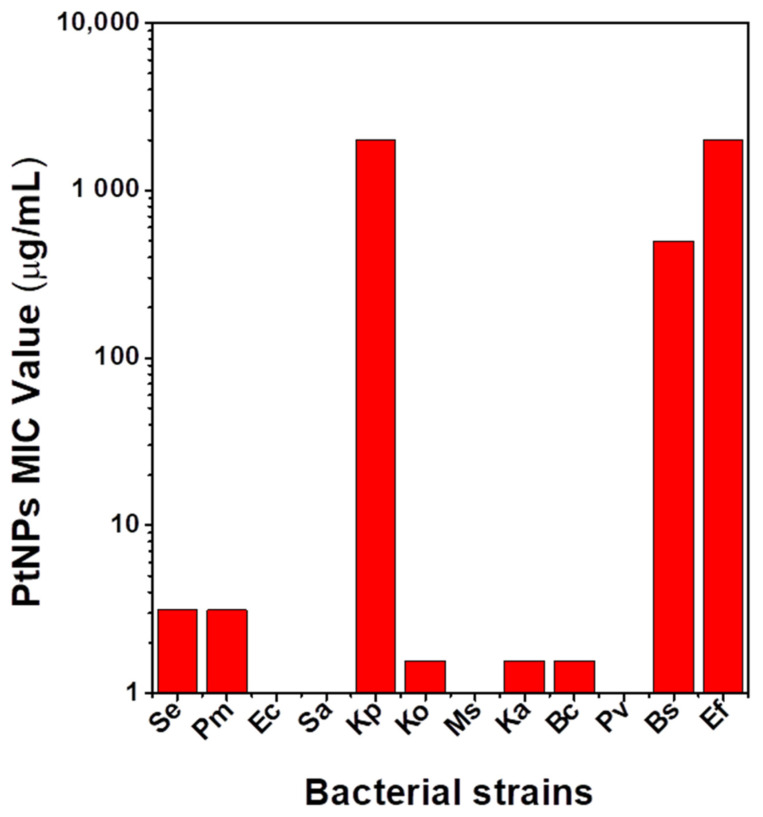
MIC values of PtNPs.

**Table 1 antibiotics-10-01275-t001:** Assignment of IR bands for CE extract and as-synthesized PtNPs.

S.N.	CE Extract Vibrational Mode (cm^−1^)	PtNPs CE Vibrational Mode (cm^−1^)	Assignment
1	3260	3283	OH- Stretching
2	2932	2920, 2850	C-H Stretching
4	1715	1708	C=O Stretching
5	1609	1620	COO- Stretching
6	1515	1505	N-O Stretching
7	1440 1328	1443 1338	C-N Stretching of Aromatic Amine
8	1174	1170	S=O Stretching
9	1052	1057	C-O-C Stretching
10	837	835	C-H Bending
11	769	759	C-H Bending

## Data Availability

The data presented in this study are available in this manuscript and Appendix A.

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
