# Peer review of "Biosynthesis of Smaller-Sized Platinum Nanoparticles Using the Leaf Extract of Combretum erythrophyllum and Its Antibacterial Activities"

_antibiotics, 2021, doi:10.3390/antibiotics10111275_

Round 1

Reviewer 1 Report

Dear authors,

I  have gone through the manuscript, and found quite interesting. I made some suggestions and comments to improve the readability of the manuscript.

  1. The 1st line of the introduction is not clear. Try to rephrase it.
  2. There are many grammatical errors and spelling mistakes.
  3. In the introduction section, novelty of the study should be emphasized and should be presented in a story format.
  4. Results and discussion section needs to be improved.
  5. The antibacterial activity section is not properly explained. It would be nice if the authors provide zone of inhibition pictures or some other graphs for showing antibacterial effect.
  6. Try to include some recent references and also some references from the Journal of antibiotics.

Author Response

 I  have gone through the manuscript, and found quite interesting. I made some suggestions and comments to improve the readability of the manuscript.

  1. The 1st line of the introduction is not clear. Try to rephrase it.

Response: It has been rephrased.  Please see lines 36-38.

2. There are many grammatical errors and spelling mistakes.

Response: The whole manuscript was sent for grammatical correction.

3. In the introduction section, novelty of the study should be emphasized and should be presented in a story format.

Response: This has been emphasized as suggested. Please See lines 71-84.

4. Results and discussion section needs to be improved.

Response: The results and discussion has been improved. Please see lines 91-93, 144-146 and 184-191.

5. The antibacterial activity section is not properly explained. It would be nice if the authors provide zone of inhibition pictures or some other graphs for showing antibacterial effect.

Response: The MIC has been presented in a graph. Please see figure  6.

6. Try to include some recent references and also some references from the Journal of antibiotics.

Response: This has been added as suggested. Please see ref. 3, 23 and 25-29.

Reviewer 2 Report

The submitted manuscript entitled “Ultra-Small Bioinspired Green Synthesis of Platinum Nanoparticles Using the leaf Extract of Combretum erythrophyllum and Its Antibacterial Activities”, although interesting has some major and a large number of minor omissions. In my opinion the work is not carefully written and some sections seem unfinished. Since the work has potential, major corrections must be made in order to be accepted.

-The title should be reformulated

-Introduction-Line 35, 36: It is not clear for what this sentence stands for: This is often due to the variation obtained in the size, shape, wavelength, surface area, and functionalisation.

Line 40: this sentence needs to be reformulated

-Line 67-68: bioreductant not bio-redundant

-Line 84: “formation of from PtNPs”, word “from” should be deleted

-Uv-Vis should be written uniformly in the text line, either as UV-Vis or Uv-Vis (line 72 vs. line 84)

-Line 84: This sentence: ”The Uv-Vis spectroscopy further confirmed this at 24 h of reaction” is not clear, it should be rewritten

Figure 1. It seems that the marks (size and font) on x- and y- axes are not equal (uniform)

-Line93: word “that” should be deleted

-Line 100: the authors refer on Scherrer equation. This equation should be either presented or referenced in the text line.

-Line 111: instead of “which agree” it should be “which are in agreement”

Figure 3. Tags in Figure caption (A,B,C,D) are not the same as tags in the Figure (a,b,c,d). Use either small or capital letters.

Figure 4. There are no (A) and (B) marks in the Figure as indicated in the Figure caption. Instead of (A) and (B) it could be written black-…and red-…Text under (B) “PtNPs synthesised CE extract.” Should be PtNPs synthesised from CE extract.” Also, marks for the superscript should be corrected (cm-1) on the Figure

-Line 133: 1620 cm−1, here -1 should be in superscript

Table 1. Table title “FTIR Wavenumber and Possible Bonds of CE Extract and its PtNPs” should be changed. Also the whole column under “Scheme 1. should be deleted and it should be written Assignment instead of Type of bond and Vibrational mode instead of Wavenumber.

Line 149 and 150: The authors say that “Staphylococcus aureus and Proteus mirabilis showed susceptibility to the CE capped PtNPs but at a higher MIC value of 3.125 μg/mL.” but in the Table 2 it is shown just for the Proteus mirabilis? Here it should be Staphylococcus epidermidis instead of Staphylococcus aureus.

That is also confirmed in Line 152-153  where it is said that:” The PtNPs showed no inhibitory effect against Escherichia coli, Staphylococcus aureus, Proteus vulgaris, and Mycobacterium smegmatis.”

Table 2. Title should be changed. It is not explained elsewhere in the text what does S/N stand for?

Moreover, in the section 2.5. Antibacterial activity Streptomycin is not mentioned at all, although there is a column (Streptomycin μg/mL) in Table 2. It is just mentioned later in the text in Materials and Methods (Line 160 and 194).

-Line 173. Sentence should be corrected.

-Line 201: synthesize not synthesise

-Line 203: bioreductant not bio-redundant

-Conclusion should be rewritten.

Author Response

1.The title should be reformulated

Response It has been reformulated. Please see the title.

2, -Introduction-Line 35, 36: It is not clear for what this sentence stands for: This is often due to the variation obtained in the size, shape, wavelength, surface area, and functionalisation.

Response: The sentence has been rephrased. Please see lines 36-38.

  1. Line 40: this sentence needs to be reformulated

Response: This has been reformulated. Please see lines 42-44.

  1. -Line 67-68: bioreductant not bio-redundant

Response: This has been corrected. Please see line 70-71.

  1. -Line 84: “formation of from PtNPs”, word “from” should be deleted

Response: It has been deleted as suggested. Please see line 91.

  1. -Uv-Vis should be written uniformly in the text line, either as UV-Vis or Uv-Vis (line 72 vs. line 84)

Response: This has been corrected throughout the text

  1. Line 84: This sentence: ”The Uv-Vis spectroscopy further confirmed this at 24 h of reaction” is not clear, it should be rewritten

Response: It has been rewritten. Please see lines 92-93.  

  1. Figure 1. It seems that the marks (size and font) on x- and y- axes are not equal (uniform)

Response: It has been adjusted accordingly.

9.-Line 93: word “that” should be deleted

Response: It has been deleted. Please see line 105.

10.Line 100: the authors refer on Scherrer equation. This equation should be either presented or referenced in the text line.

Response: It has been referenced. Please see ref. 25.

11.Line 111: instead of “which agree” it should be “which are in agreement”

Response: It has been rewritten. Please see line 112.

  1. Figure 3. Tags in Figure caption (A,B,C,D) are not the same as tags in the Figure (a,b,c,d). Use either small or capital letters.

Response: Figure 3 has been adjusted accordingly.

13 Figure 4. There are no (A) and (B) marks in the Figure as indicated in the Figure caption. Instead of (A) and (B) it could be written black-…and red-…Text under (B) “PtNPs synthesised CE extract.” Should be PtNPs synthesised from CE extract.” Also, marks for the superscript should be corrected (cm-1) on the Figure

Response: Captions of (a) and (b) has been included for ease of reference and the superscript included

14 Line 133: 1620 cm−1, here -1 should be in superscript

Response: It  has been written in superscript. Please see line 149.

  1. Table 1. Table title “FTIR Wavenumber and Possible Bonds of CE Extract and its PtNPs” should be changed. Also the whole column under “Scheme 1. should be deleted and it should be written Assignment instead of Type of bond and Vibrational mode instead of Wavenumber.

Response: All has been addressed as suggested.

  1. Line 149 and 150: The authors say that “Staphylococcus aureus and Proteus mirabilis showed susceptibility to the CE capped PtNPs but at a higher MIC value of 3.125 μg/mL.” but in the Table 2 it is shown just for the Proteus mirabilis? Here it should be Staphylococcus epidermidis instead ofStaphylococcus aureus. That is also confirmed in Line 152-153  where it is said that:” The PtNPs showed no inhibitory effect against Escherichia coliStaphylococcus aureus, Proteus vulgaris, and Mycobacterium smegmatis.”

Response: It has been corrected accordingly. Please  see line 177.

  1. Table 2. Title should be changed. It is not explained elsewhere in the text what does S/N stand for? Moreover, in the section 2.5. Antibacterial activity Streptomycin is not mentioned at all, although there is a column (Streptomycin μg/mL) in Table 2. It is just mentioned later in the text in Materials and Methods (Line 160 and 194).

Response: The table title has been changed. It is now in the supporting document.  S.N.  means “serial  number”. 

  1. Line 173. Sentence should be corrected.

Response: It has been corrected. Please see lines 209-210.

19.Line 201: synthesize not synthesise

Response: It has been corrected. Please see line 240.

  1. Line 203: bioreductant not bio-redundant

Response: It has been corrected. Please see line 242.

  1. Conclusion should be rewritten.

Response: The conclusion has been rewritten. Please see lines 245-246 and 248-249.

Reviewer 3 Report

The submitted manuscript studied the green synthesis of platinum nanoparticles using the leaf extract of Combretum erythrophyllum and tested its antibacterial activities. Despite the exciting topic and nice and clear Introduction, the manuscript lacks the crucial aspects that the manuscript should have. First of all, it is the novelty of the subject.  The authors themselves provide several references on the identical (very similar) topic. The only difference is the use of the different plants. However, it is not clear the advantage of using Combretum erythrophyllum compared to other plant extracts. Moreover, the Results are insufficiently presented. TEM images in Figure 3 are of poor quality, and Figure 3d is unreadable. The purpose of Figure 4 and Table 1 is unclear. But the most important, the flagship of the manuscript – antibacterial activity, is presented only very weakly.  Table 2 is chaotic and it is not clear what the conclusion is. The discussion of results is insufficient and lacks comparison with other studies and a description of why this combination is worth study. But it definitely needs better results visualisation. I am sorry, but I can not recommend a manuscript for publication in this form.   

Author Response

  1. First of all, it is the novelty of the subject.  The authors themselves provide several references on the identical (very similar) topic. The only difference is the use of the different plants. However, it is not clear the advantage of using Combretum erythrophyllum compared to other plant extracts.

 Response. Please see lines 71-84  for the novelty of the work.

  1. TEM images in Figure 3 are of poor quality, and Figure 3d is unreadable.

Response: Better images have been provided. Please see figures 3 and 4.

  1. The purpose of Figure 4 and Table 1 is unclear.

Response: Figure shows the FTIR spectra of both the CE extract and its as-synthesized PtNPs. They are discussed in section 2.4. The table shows the assignment of the bands for easy identification.

  1. But the most important, the flagship of the manuscript – antibacterial activity, is presented only very weakly. 

Response: Discussions on the antibacterial activity has been improved, please see lines 184-191.

  1. Table 2 is chaotic and it is not clear what the conclusion is.

Response: Table 2 has been presented in graphical format for ease of reference.

  1. The discussion of results is insufficient and lacks comparison with other studies and a description of why this combination is worth study.

Response: The results obtained in this study were compared with other studies. Please see lines 97-98 and  108-109.

  1. But it definitely needs better results visualisation. I am sorry, but I can not recommend a manuscript for publication in this form.   

Response: The manuscript has been improved based on all the suggestions.

Round 2

Reviewer 1 Report

Happy to see the improvisation of the manuscript.

Author Response

Thank you for accepting the improved manusript.

Reviewer 2 Report

In the revised Manuscript, Dr. Oluwafemi et al. responded to all given requests.  

However, some minor corrections still need to be made and I recommend that the authors go through the whole Manuscript thoroughly. I also noticed that the Manuscript title given in the supplementary file is not the same as in the revised Manuscript. Besides, it is not clear why this table was removed from the Manuscript and transferred to Supporting Document? In my opinion the table title is still not appropriate.

Author Response

  1. In the revised Manuscript, Dr. Oluwafemi et al. responded to all given requests.  However, some minor corrections still need to be made and I recommend that the authors go through the whole Manuscript thoroughly.

Response: Thank you for the suggestion; the entire manuscript has been proofread and corrected.

  1. I also noticed that the Manuscript title given in the supplementary file is not the same as in the revised Manuscript.

Response: Thank you for the comment; the supplementary title has been adjusted accordingly.

  1. Besides, it is not clear why this table was removed from the Manuscript and transferred to Supporting Document? In my opinion the table title is still not appropriate.

Response: Thank you for the comment; table S1 has been given a more appropriate title. Based on the suggestion of one of the reviewers, the table was moved to the supporting document. Also, the MIC data of the PtNPs is now graphically presented.

Reviewer 3 Report

Dear authors,

I appreciate your improvements in the manuscript. However, I am still missing the three crucial pieces of information:

  • Why are your particles so small compared to references 2 and 23?
  • How do you know that in Figure 3, you see the Pt nanoparticles and not Ca or Cu nanoparticles, as your EDX spectra suggest?
  • Why did you obtain such different results in Klebsiella species (Kp vs. Ko and Ka)? Why does the same principle of oxidative stress do not apply in other species, as you discuss on lines 182-189? Moreover, the newly added part (182-189) is very speculative and must be improved. Antibacterial activity of your nanoparticles is the main goal of the manuscript and must definitely contain more information and comparison than is present.

Author Response

I appreciate your improvements in the manuscript. However, I am still missing the three crucial pieces of information:

  1. Why are your particles so small compared to references 2 and 23?

Response: This is due to the different plants used as a bio-reductant which contains different biomolecules that are responsible for the reduction and stabilization of the PtNPs. In addition, different procedures were used for the biosynthesis of the nanoparticles.

2.How do you know that in Figure 3, you see the Pt nanoparticles and not Ca or Cu nanoparticles, as your EDX spectra suggest?

Response: Thank you for the comment. However, as stated in the manuscript, the Cu is from the Copper grid used for the TEM analysis, and the Ca is from the plant. Also, no copper or calcium salts precursor was used in the synthesis, and the platinum salt was 99.99% pure. In addition, the selected area electron diffraction (SAED) rings correlated to the XRD patterns confirming the formation of PtNPs. Please see lines 119-121 and 126 -128. Furthermore, the absorption spectrum did not show any peak related to Cu or Ca.

  1. Why did you obtain such different results in Klebsiella species (Kp vs. Ko and Ka)? Why does the same principle of oxidative stress do not apply in other species, as you discuss on lines 182-189? Moreover, the newly added part (182-189) is very speculative and must be improved. Antibacterial activity of your nanoparticles is the main goal of the manuscript and must definitely contain more information and comparison than is present.

Response:  Thank you for the comment; the different result obtained for Klebsiella pneumoniae is because of its more virulent and resistant nature than other Klebsiella species.  Please see lines 182-188 for more explanation. The previously added section, as requested, was proposed based on previous findings of the cited ref. 27, 30, and 31.